# The Role of Farmers' Umbrella Organizations in Building Transformative Capacity around Grassroots Innovations in Rural Agri-Food Systems in Guatemala

Rosalba Ortiz [1,*] and Jordi Peris [2]

1 Doctoral School, Universitat Politècnica de València (UPV), Calle Universidad, 46003 Valencia, Spain
2 Project Engineering Department, Universitat Politècnica de València (UPV), Calle Universidad, 46003 Valencia, Spain; jperisb@dpi.upv.es
* Correspondence: roorval@doctor.upv.es

**Abstract:** Family farmers and grassroots innovations can enable transitions to more sustainable food systems. The study explores the roles umbrella farmers' organizations play in building transformative capacity through grassroots innovations in rural food systems in Guatemala. An analytical framework based on the notion of transformative capacity and socio-technical transitions is used to explore the specific factors enabling and limiting the transformative potential of grassroots innovations in a rural setting. A case study in rural Huehuetenango, Guatemala is presented, using interviews and focus groups discussions with relevant stakeholders engaged in the development process. Perceptions from interviews and focus groups discussions highlight the catalyst role played by the umbrella farmers' organization as the main enabling factor to increase transformative capacity of grassroots innovations. The umbrella organization plays a key role in enabling farmers to pursue socio-technical transformations and in moving grassroots innovations outside a niche sphere. It contributes to creating coherence towards a common sustainability vision, supporting innovation and experimentation, and providing technical assistance around core development processes. In addition, it navigates across different levels of agency (households, communities, networks, and institutions) and different interaction scales (local, department, and national). However, gender and multi-generational gaps have been identified as limiting factors that would require further analysis.

**Keywords:** transformative capacity; agri-food systems; rural Guatemala; grassroots innovations; niche innovations; umbrella farmer organizations; family farmers

## 1. Introduction

Producing more food, while building resilient agri-food systems requires socio-ecological innovations, as well as changes in how institutions and stakeholders organize and operate [1–4]. Increased academic attention is paid to the factors that lead to sustainable transitions [5–8] particularly in agri-food systems [9–14].

There is an urgent need to move towards more sustainable food production [1–3], and such transformation requires active engagement of all relevant stakeholders [4,5]. Social innovations may define sustainable agendas, change institutions and attitudes towards increased resilience and improved livelihoods [6], and different governance forms [7,8]. Many solutions will be bottom-up or grassroots innovations [9].

Family farmers and grassroots innovations can play a crucial role in enabling transitions to more sustainable societies [2,10] and resilient food systems [11–14]. Grassroots innovations are usually carried out by groups operating outside mainstream innovation processes [15]. Grassroots innovation movements are important because they have the capacity to empower local communities to foster change [16]. The transformative potential of grassroots innovations depends on the power of actors and their networks in challenging systemic changes [7,17]. Therefore, greater participation in transitions is required [18,19]

for grassroots innovations to overcome niche experimental status. In the multi-level perspective, niches are the location where radical innovations are developed and may have the potential to change regime practices [20]. Grassroots innovations are perceived as domains of transformation [21], in which changes depend on internal tensions within the regime and on the development and adaptive processes of the niche innovations [17,19]. Still, many grassroots innovations in family farming continue as niches, unable to challenge existing norms and institutions in the regime [22–24].

Some research has been carried out on the role of umbrella or second-tier farmers' organizations and social movements in creating identity and momentum for transformations of grassroots innovations [25,26]; innovations that are often overlooked by formal institutions [27]. However, the presence of an umbrella or second-tier farmer organizations with its own identity [28] and appropriate assistance may generate powerful change over time [29].

By umbrella or second-tier organization, we understand a network or coalition of farmer organizations, which builds identity among a group of organizations to solve social problems together [25]. Umbrella farmers' organizations can have advocacy and representative functions, they move local agendas of agricultural producers, allowing access to buyers, financial services, and certification schemes [28]. Yet, the role of civil society, culture, and social movements in transitions needs further analysis [30]. Further research is needed on the roles of local actors in strengthening the transformative capacity of niche innovations to overcome experimental status [31] and to challenge existing regimes. This paper addresses this research gap, by studying the role of umbrella farmer organizations in building transformative capacity of grassroots innovations to overcome niche spheres.

The analysis of what triggers transformative capacity should consider domains of transformation influencing the governance within niche innovations as suggested by Andersen et al. [21] and relations between niche, regime, and landscape levels [32] governing regimes and power relations for the transitions as suggested by Avelino et al. [7] and Smith and Stirling [27], furthermore looking into different bridging roles that stakeholders can play to scale-up transformation from grassroots innovations [8].

The paper is structured as follows: Section 2 addresses the conceptualization of resilient agri-food systems and the literature review to justify the use of the different factors influencing the transformative capacity for resilient transformation in agri-food systems in a rural setting; Section 3 describes the methods and the case study. Section 4 presents the main findings, while Section 5 presents the discussion, and Section 6 summarises the main conclusions.

## 2. Multi-Level Perspective and Transformative Capacity in Agri-Food Systems

Agri-food systems entail different actors and activities involved in the production, processing, distribution, consumption, and disposal of food products that originate from agriculture [33]. It entails a network of actors and activities interacting in a social, cultural, political, and ecological context [34–36]. Agri-food systems, as any other socio-ecological system, are characterized by interconnections, mutual dependencies, and dynamic relationships between humans and the environment [37–39]. The combination of different types of knowledge and systems' thinking and learning is critical to resilient transitions [23,38,40]. A resilient agri-food system should have the capacity to adapt and transform itself so it can persist in the long-term [41], learning to live with change and uncertainty [38].

The multiple level perspectives (MLP) in the socio-technical transitions framework [20,32] has been used to explain how innovation and changes happen at different scales, from local to regime and landscape levels [42] (Figure 1).

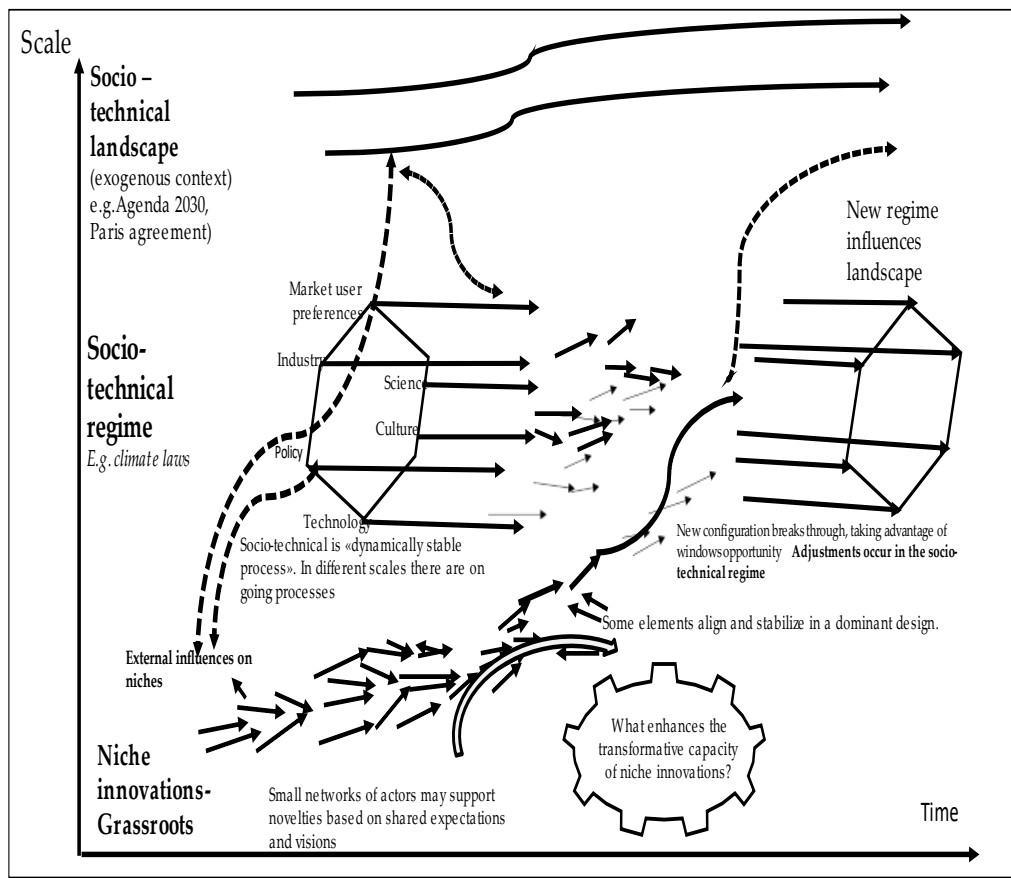

**Figure 1.** Multi-level perspective on transitions. Source: adapted from Geels and Schot [43].

The MLP allows explorations on how "niche" or grassroots innovations lead to changes in the system [8,44–46]. Rules in the regime can be seen as constraints, but they also enable relevant stakeholders to modify or replace existing rules [47]. The new rules might eventually build trust and reliability [48,49]. Overall, the MLP embraces the sense of multiple agents interacting, which leaves space for different actions [42]. In this sense, transition research seeks to understand how different types and forms of agency influence transitions, and how they engage to reach the desired transitions more effectively, including the participation of marginalized groups [50] and the roles that different stakeholders can play to strengthen the transformative capacity of grassroots innovations [51–53].

It is argued that MLP can be strengthened by emphasising governance issues, and by engaging not only decision makers, but also civil society organizations and other relevant stakeholders [21,46]. The conceptualization of transformative capacity is then crucial to understand how transition pathways are initiated, realized, or contested at niche, regime, and landscape levels.

- Transformative capacity in family farming

The idea of transformative capacity originates from research on resilience theory and socio-ecological systems [46]. The analysis of transformative capacity in family farming requires the understanding of the interdependencies between ecological and social processes [54].

Departing from the existing literature on socio-technical and socio-ecological transitions, Wolfram [55] developed a comprehensive framework with several components defining the transformative capacity of niche innovations. Transformative capacity is defined for an urban context as:

*"The collective ability of the stakeholders involved in urban development to conceive of, prepare for, initiate and perform path-deviant change towards sustainability within and across multiple complex systems that constitute the cities they relate to."*

Wolfram [55] identified a set of 10 interdependent components (C), which define the transformative capacity in a territory. Components C1 to C3 refer to agency and interaction forms; while C4 to C7 identify core development processes such as sustainability and system awareness, knowledge, and embedding; C8 addresses knowledge and reflexive learning; and C9–C10 represent relational dimensions affecting the rest of the components. C9 considers different levels of agency: individual, households, institutions, while C10 considers different scales: local, regional, national (Figure 2).

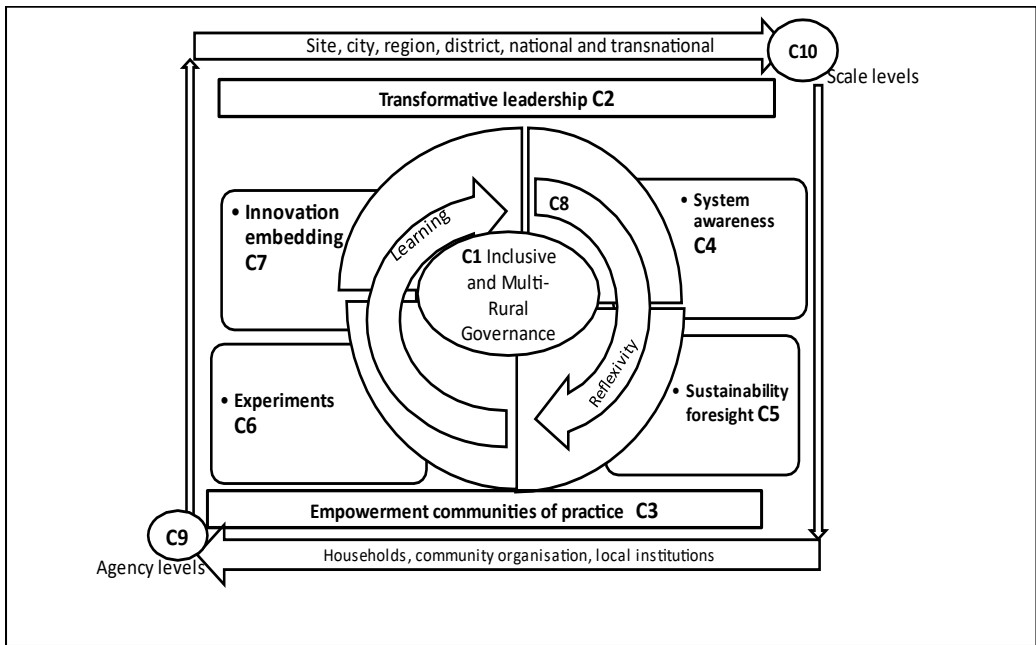

**Figure 2.** Independent components of transformative capacity. Source: Wolfram [55].

To make an analogy with Wolfram's [55] framework for the analysis of the transformative capacity in a rural area, it is imperative to justify it with relevant literature about transformative capacity related to agri-food systems in rural contexts.

### 2.1. Transformative Capacity and Agency Interaction Forms (C1–C3)

- Inclusive and multi-level governance (C1) encompasses participation and inclusiveness, governance [41] network forms, and sustained intermediaries and hybridization [55]. Sustainability transitions research acknowledges the importance of who governs, whose system counts, and whose sustainability has priority [56], as well as the inherent power relations a food system entails [18,34,37]. Bui et al. [57] highlight the crucial role of local authorities in regime reconfiguration. Some transformative processes can put participants at risk, especially if the new ideas change power relations. In this context, bottom-up governance may enable key bridging elements, such as, shared vision, networking, knowledge creation, resource provision, and conflict resolution [8].
- Transformative leadership (C2) is also considered a key factor in enhancing the transformative capacity of niche innovations. Actors in leading positions can define the initial phases of transformative change [33,58]; leaders who often have wide networks, are charismatic, have legitimacy [19], and play key roles in building leadership around shared visions [8,57]. Transformative leadership can promote a sustainability agenda in rural agri-food systems.

- Empowered communities of practice (C3), in which social needs and motives are considered, and deficits are addressed by public policies [55]. Promoting innovation for, by, and with smallholders is key for the transformation towards resilient food systems [40,56]. Rural transformation is triggered by behavioral changes because of new knowledge in local living labs and experiments [59] and empowered communities of practice [18]; as well as mediation between different knowledge sources: farmer to farmer, institutional, and private actors. Diversity of grassroots experimentation in terms of initiatives, technologies, and demands, and its complexity often challenge the willingness of mainstream institutions to adapt to novel ideas [15].

## 2.2. Transformative Capacity and Core Capacity Development Processes (C4–C7)

- System awareness (C4): The importance of agricultural diversity in strengthening resilience is recognized [40,51,52,60]. Attention is put on the meaning of the territory for a resilient agri-food system and the importance of the reconnections between agriculture, food, and the environment [12,14,40,53]. A systemic vision is considered to stimulate transitions [61]. The system provides a comprehensive view on actors and factors that co-determine innovation, which allows a better understanding of complexities in the food system [58].
- Shared sustainability foresight (C5): The need of a shared vision in agri-food systems and its alignment with the demands of relevant stakeholders has been highlighted [19,21,57], in which inclusive and participatory planning play a key role [40]. Sustainability foresight offers an avenue for the creation of new actor networks; and the creation of concrete strategies with a high chance of implementation [19].
- Diversity in community-based experimentation and innovation (C6): wider exposure of diverse stakeholders to experimentation and innovation can strengthen local economies and socio-ecological resilience in rural settings, if local economies are exposed to a wider range of strategies [62]. Niche development and interactions are key processes in transitions leading to the integration of new rules and practices into profound regime reconfigurations [11] and open spaces for emergent opportunities, enabling new actors to engage and novel practices to evolve [63]. Engagement, inclusion, and participatory approaches in transitions are, therefore, crucial [18–21].
- Innovation embedding and coupling (C7): innovations encompass niche innovations organized around a small network of actors sharing the will to break away the dominant regime [11,61]. It includes anchoring and linkages to the regime [54,64,65]. Developing new visions of farming and food is a key aspect of niche development [11,66,67]. The reflexive relationships between network actors and their institutional environment in which they are embedded is highlighted by Klerkx et al. [58]. Trust plays a key role in building institutional networking and collaboration [68]. Ultimately, maintaining livelihoods has much to do with learning, adaptation, and alignment [40].

## 2.3. Transformative Capacity and Learning and Reflexivity (C8)

Farmers' knowledge and multi-actor knowledge networks that facilitate exchanges and joint learning are crucial for resilience agriculture; farmers need to re-learn and change mind sets to disrupt with mainstream unsustainable practices [17,18]. Participatory knowledge processes demand network, team building, and openness [56]. It entails how learning is enhanced by experimentation and networking in building transformative capacity around social innovations [40,66], as part of social interaction and reflexive learning [17,64]. Eventually, it will enhance farmers' capacity to cope and to deal with uncertainties through learning and reorganization [18] and strengthen the social structures through which practices are disseminated [68]. Transformative social innovations should focus on learning, knowledge creation, and on empowering people involved, e.g., marginalized communities to eventually become agents and drivers of social innovation [64].

### 2.4. Transformative Capacity and Relational Representations (C9–10)

Different levels of agency (C9): interactions vary according to the level of agreement between grassroots and institutions, and such interactions are always changing and can even co-exist for short periods of time as they transition from one relationship to another [24]. Agency clearly plays a role at each stage of the process [41,69].

Different scales (C10): innovation ecosystems thinking should investigate transboundary linkages between sectors and promote enabling environments to better address cross-cutting sustainability issues and alternative approaches to agriculture [23]. Interactions between niche actors and local authorities, and at different levels, farm, local community, regional, and national, are recommended [41,70].

The literature review presented in this Section 2 supports the analogy with Wolfram's [55] heuristic framework to understand factors enhancing the transformative capacity of agri-food systems in a rural context. The set of components explained in Section 2.1 was used to prepare the guiding questions for the interviews carried out for the case study. The following definition of transformative capacity was assumed for a rural context: "The collective ability of the stakeholders involved in rural development to envision, prepare for, and promote changes towards sustainability in family farming systems in a rural territory". Adapted from Wolfram [55].

## 3. Methods

### 3.1. Qualitative Analysis

The analysis is qualitative, based on an interpretative paradigm that entails a close interaction with targeted subjects [71,72] to better understand their perceptions [73,74]. A qualitative analysis contributes to get context- and culturally anchored perceptions and knowledge [75,76]. The study carried out individual interviews, focus groups discussions and document analysis as main sources of information.

### 3.1.1. Stakeholders Interviewed

Primary information comes from interviewed subjects, relevant stakeholders who are engaged in grassroots innovations, such as Climate Adapted Villages (CAV). CAV is one of the main grassroots innovations promoted by second-tier farmers' organizations among family farmers in rural Huehuetenango, Guatemala. The first list of interviewees included eight persons, who represented relevant institutions interacting with farmers in the area: from farmer organizations, public institutions to private sector. The first list of interview subjects was recommended by donors and technical people working in the second-tier farmer organization. This list was later increased to 14, based on recommendations by the first interviewees. Farmers and women farmers, private sector, government, and local and international donors are among stakeholders included (Table 1).

We reached a saturation point [77,78] when novel information became scarce towards the last interviews. Most of the interviews were carried out face-to-face in Huehuetenango in 2020. All interviewed subjects signed a paper or electronic copy of prior consent for their participation in this study.

Male and female farmers, private sector, government representatives, and local and international donors are among stakeholders included (Table 1).

**Table 1.** Interviews in Huehuetenango Guatemala: List of interviewed stakeholders.

| ID | Affiliations | Scale |
|---|---|---|
| | Government (Go) | |
| Go1 | Ministry of Agriculture—MAGA (Municipal level and Department level) | Local, department, national |
| Go2 | Instituto Nacional de Bosques—INAB (Municipal and Department level) | Local, department, national |
| | | Local, department, national |
| Go3 | SEGEPLAN—Secretaria de Gobernación y Planificación (Department level) | Local; department, national |
| Go4 | Consejo Nacional de Areas Protegidas—CONAP (Department level) | Local, department, national |
| Go5 | Municipal authorities in San Miguel Acatán. Comité Municipal de desarrollo (COMUDE) | Local |
| | Farmers' organizations | |
| UFO | UFO1- Director Asociación de Desarrollo de los Cuchumatanes (ASOCUCH) Network of farmers<br>UFO 2. Expert on governance and public policies<br>UFO 3. Expert on gender<br>UFO 4. Expert on family farming and food security | Local, department, national |
| FO1 | Female leader in Cooperative ASMADI—CBO member of ASOCUCH | Local |
| | Donors—(Do) | |
| Do1 | Rainforest Alliance-USAID funding | Local/international NGO |
| Do2 | Helvetas-Swiss cooperation | |
| Do3 | Nexos Locales-USAID funding | |
| Do4 | The Development Fund Norway—Norwegian Development cooperation (Norad)-funding | International NGO |
| | ALLIANCES-collaborative (ALi) | |
| ALi 1 | COFETARN representative -Member of COCODE | Network |
| Ali 2 | Mesa departamental de cambio climático | Network |
| | Private sector (Pri) | |
| Pri1 | Cámara de la Miel | Network |
| | Focus group (FoG) | |
| FoG1 | Technical experts | Local |
| FoG2 | Female farmers | Local |

Source: own elaboration.

### 3.1.2. Focus Groups Discussions

Two focus groups were organized: one involving technical experts in family farming to validate the role of different institutions in relation to components and sub-components of transformative capacity, and with female farmers to assess their level of participation in grassroots innovations.

Focus group 1 with technical experts. A focus group was organized among technical people from the second-tier organizations and some technical people working with donors to discuss and validate the role of different organizations in moving transitions outside the niche sphere. These were experts working on development issues among family farmers in rural Huehuetenango. In total, eight people participated (four women and four men). Participants were selected among those who previously participated in individual in-depth interviews. In this sense, participants in the focus group already had some understanding of the topic under analysis that enriched the discussions during the focus group sessions. The discussion was centred around the roles that different stakeholders play in strengthening the transformative capacity of grassroots innovations in order to jointly find out who had a prominent role in strengthening the transformative capacity of grassroots innovations for family farmers in rural Huehuetenango. Topics discussed were divided into categories based on the components that explain transformative capacity. Some categories were combined to facilitate the discussion (Table 2).

**Table 2.** Focus group 1 technical experts: Categories and subcategories discussed.

| Focus Group 1 with Technical Experts | |
|---|---|
| **Main Categories** | **Sub-Categories** |
| System awareness (C4) and sustainability (C5) | Family farming' system<br>Shared sustainability vision |
| Stakeholders' role (C1)—-governance | Inclusive<br>Collaborative |
| Leadership (C2) | Transformative<br>Inclusive<br>Power (elite capture) |
| Experiments (C6) | Grassroot innovations<br>CAV |
| Empower communities of practices (C3) | Capacity development<br>CAV |
| Learning and reflexivity (C8) | Participatory<br>Continuity in technical support<br>Reflections and feedback |
| Agency level (C9) | Households<br>Community<br>Institutions |
| Innovation embedding (C7) and Scale levels (C10) | Local<br>District<br>National<br>International |

Source: own elaboration.

Focus group 2 consisted of female farmers. It involved 10 female farmers, all members of local associations, cooperatives, and women's groups. The focus group focused on discussing women's participation in grassroots innovations (using CAV as an example). The discussion considered participation in three main categories: women's participation in activities, decision-making, and capacity-development (Table 3). Since these women are already organized, they may have more opportunities to participate than those who are not organized at all. However, such comparison was not part of this study.

**Table 3.** Focus group 2 with female farmers:Categories and sub-categories discussed.

| Main Categories | Sub-Categories |
|---|---|
| Women's participation in CAV activities | Planning<br>Green micro-credits<br>Collaborative activities in CAV |
| Women's participation in decision-making | Participation in committees in local organizations<br>Participation in boards |
| Women's participation in capacity-development | Tailored technical assistance<br>Tailored training |

Source: own elaboration.

### 3.1.3. Documental Analysis

The information was triangulated with the analysis of policies governing initiatives towards the resilience and sustainability of rural family farming in Guatemala.

Document analysis: international policies, national strategies, and actions plans for agriculture, climate, and resilient agri-food systems were analysed to understand the existing regime and landscapes, which may influence transitions towards sustainability in family farming in rural areas of Guatemala. Document analysis was also used to contrast

perceptions gathered through the interviews and the focus group discussion This analysis contributed to isolate the contribution of the umbrella organization creating condition for CAV to transit from local, to department and national levels (Table 4).

**Table 4.** List of documents analysed.

| Tittle of the Document | Type of Document | Level of Impact | | |
|---|---|---|---|---|
| | | National | Department Level | Local Districts (Micro-Watershed) |
| "Kat'un 2032—Our Guatemala" | Guatemalan National development strategy towards 2032 | X | X | X |
| Ley de Seguridad Alimentaria y Nutricional -Ley SAN | Law on food and nutritional security | X | X | X |
| Ley Marco de Cambio Climático (Decreto 7-2013) | Law, on climate change and resilience | X | | |
| Ley de Incentivos Forestales Para Pequeños Poseedores de Tierra- Ley PINPEP | Law on forest incentives for smallholders' forest owners (also communal lands) | X | X | X |
| ASOCUCH upscales CAV as a new scheme for ecosystems conservation and resilient food production (PINPEB + climate law regulations) | | | | |
| Plan estratégico 2019–2023. Asociación de Desarrollo de los Cuchumatanes | Strategic plan for second-tier umbrella farmer organization | | X | X |
| CAV is integrated in ASOCUCH's strategic plan and local farmers organizations' action plans | | | | |
| Climate adapted villages (CAV) | Grassroots innovation (CAV) | | | X |
| CAV Pepajau, San Juan Itxcoy | Adaptation plan | | | X |
| CAV Magdalena, Chiantla | Adaptation plan | | | X |
| CAV Paijala, Sta Eulalia | Adaptation plan | | | X |
| CAV, Limón Bajo, Todos Santos | Adaptation plan | | | X |
| CAV Secheu, Concepción | Adaptation plan | | | X |
| CAV Mitlaj Chiantla | Adaptation plan | | | X |
| CAV Tojchim, Chiantala Aguacatán | Adaptation plan | | | X |
| CAV Arroyo Carpintero, Chiantla | Adaptation plan | | | X |
| CAV San Francisco, Chiantla | Adaptation plan | | | X |
| CAV El Rosario, San Migual Acatán | Adaptation plan | | | X |
| CAV Chenxul, San Rafael | Adaptation plan | | | X |

Source: own elaboration. Note: X denotes the sphere of influence: from district to national level.

*3.2. The Case Study for This Article*

The case study was carried out in rural communities of Huehuetenango, Guatemala. Huehuetenango is one of the 22 departments of Guatemala; located in the western highlands that borders with the Mexican state of Chiapas in the North and West. Huehuetenango's is one of the most diverse regions in terms of Mayan ethnic groups. Q'anjob'al, Chuj, Jakaltek, Tektik, Awakatek, Chalchitek, Akatek, and K'iche' are the predominant ethnic groups; each one with its own language [79]. Subsistence farming is predominant, and it is carried out in small plots usually smaller than 2 hectares, in the temperate climates of the Cuchumatanes Mountains situated between 2000 and 3000 m above sea level.

- Climate adapted villages a grassroots innovation

Climate adapted villages (CAV) is one of the grassroots innovations promoted by ASOCUCH, the umbrella farmers' organization, in family farming communities of Huehuetenango, Guatemala. CAV promotes collaboration between communities in a micro-watershed to work together for the implementation of an adaptation plan. The adaptation plan builds the resilience of the farming system in a selected micro-watershed. CAV planning also considers surrounding forest and water ecosystems. Using micro-watersheds as a unit of planning is not new, but the novelty of CAV is to put adaptations funds on the hands of local farmer cooperatives and associations, and in making communities protagonists in managing and monitoring the adaptation plans. Adaptations funds are used as one-time investments (e.g., building community seed banks) and part of the funds are used as green micro credits to sustain the adaptation plans. It is considered a green micro fund, because to get funding for agricultural activities, farmers pay "green interest rates" in addition to paying back the loans. Green interest rates are defined as in-kind costs of environmental improvements that farmers are committed to implement in their farms, e.g., soil conservation practices, agroforestry, seed, and biodiversity conservation activities [80].

CAV is a grassroots innovation that builds long-term resilience in rural family farming systems: it takes into consideration the local impacts of extreme weather variability on farming communities. This grassroots innovation has been implemented by The Norwegian Development Fund in Guatemala, Somalia, Ethiopia, Malawi, and Nepal. In Guatemala, other donors provided initial funding for the implementation of CAV in various communities of Huehuetenango. Funds were also matched by ASOCUCH with governmental forests' incentives that are paid to family farmers in Guatemala. CAV is currently implemented in 11 different micro-watersheds (Figure 3).

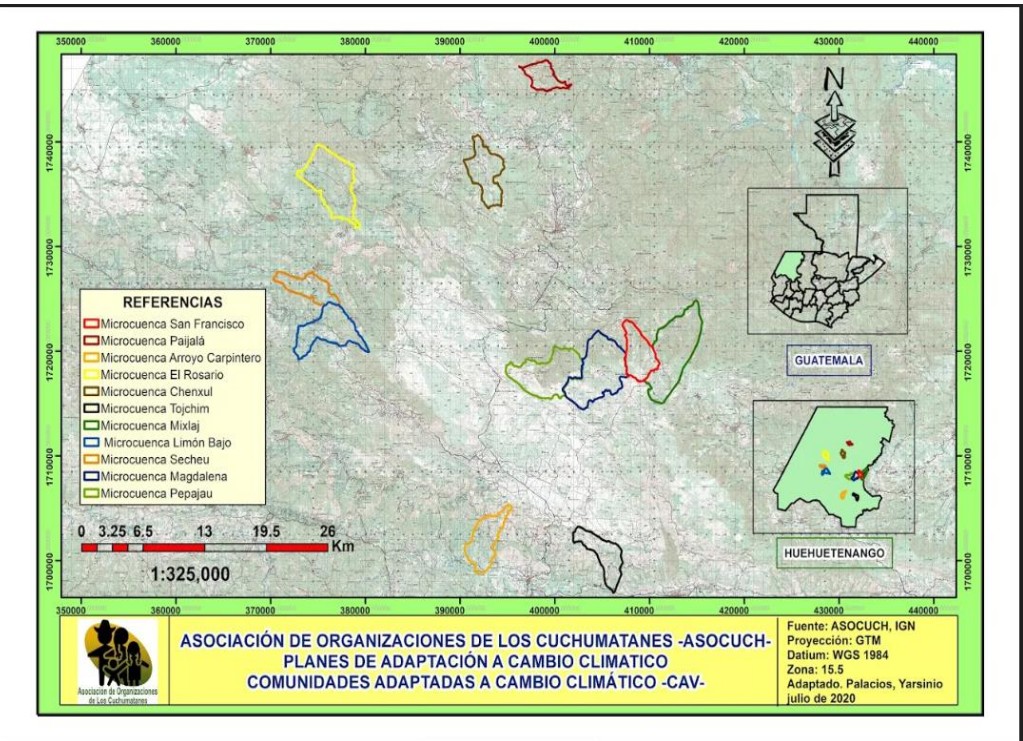

**Figure 3.** Climate adapted villages in Huehuetenango: 11 micro watersheds. Source: ASOCUCH, 2020.

CAV is built around three basic processes: knowing, doing, and sustaining. Knowing puts emphasis on knowledge, learning, and ecosystem awareness by using diverse Participatory Vulnerability Assessment (PVA) tools. The PVA is carried-out in a micro-watershed, with the participation of more than three communities each, done in a participatory way

by involving relevant stakeholders in the territory. The PVA analysis contributes to a better understanding of the socio-ecological systems in the micro-watershed, building on the knowledge, perspectives, and experiences of different stakeholders in the community. The latest scientific knowledge on local climate effects and climatic information is used to contrast and to supplement traditional knowledge. The PVA strengthens a collective understanding of changes in local climates and on the immediate and long-term impacts in local agri-food systems. It also contributes to a collective reflection on feasible measures that can be selected to reduce negative impacts as analysed and identified by the communities.

Doing focuses on planning and governance issues. During this process communities prioritise measures that will be implemented to enhance the resilience of local agri-food systems. Such measures respond to the needs identified during the PVA. Participant communities must reach a consensus on how to use the funds, who will be responsible for the various tasks, and for the management of the funds. Some power relations and conflictive situations may arise and should be resolved. Since the initial "seed" fund is unlikely to be adequate for covering all measures identified during the PVA, the communities need to mobilize other resources to fill those gaps. Steering committees were formed with representation from different communities to oversee the implementation of measures and plans, and to design advocacy and networking required.

To sustain puts emphasis on sustainability foresight, and on the means to reach it, which is often related to access to resources such as funding and relevant staff in the communities. In CAV, resources can be raised through voluntary contribution, external donors, revolving fund mechanisms, or government budgets. As donors only provide initial funding, the communities are challenged to devise a plan for how they will be able to sustain adaptation measures in the future [80].

The umbrella organization: The Asociación de Desarrollo de los Cuchumatanes (ASOCUCH) is a network of 20 rural smallholder farmers' organizations, legally registered in 2000. Its members are registered either as associations, cooperatives or women groups coming from 12 out of 31 districts in Huehuetenango. ASOCUCH has over 10,000 members, belonging to six different Mayan indigenous groups and mestizos living in the Cuchumatanes mountains. ASOCUCH has the mission to become an engine of territorial initiatives for environmental and productive management that builds local capacities for the well-being of rural families, through sustainable, equitable, and inclusive rural development [81]. Some of ASOCUCH's member organizations participate in commercial networks, such as the honey, coffee, and potato chambers, which are "public-private initiatives" coordinated by the Ministry of Agriculture (MAGA). This allows ASOCUCH's member organizations to influence the local and national agendas and to mobilise resources for capacity-building and activities towards building resilient agri-food systems.

International landscape influencing agri-food systems' regime in Guatemala: The Agenda 2030 and the Paris Climate Agreement are part of the international landscape influencing sustainability agendas in Guatemala. Guatemala embraced Agenda 2030 through its national development plan "Kat'un 2032—Our Guatemala". Kat'un 2032 calls for a transition from the current unsustainable development model to one of sustainable human development [79]. Building sustainable agri-food systems is a strategic priority in Kat'un 2032, in which food security and gender inclusion are core priorities to transit to resilient and sustainable agri-food systems in the rural areas of Guatemala. Política de Seguridad Alimentaria y Nutricional [82] acknowledges the need for locally produced nutritious food and envisages diversified agri-food systems to ensure quality and nutritious food in rural Guatemala. Guatemalan climate law [83] encourages productive practices to adapt to a changing climate, which consider traditional and ancestral knowledge, and appropriate technologies for the ecological conditions in the territories. The forest and agroforestry incentives program for smallholders, Programa de incentivos para pequeños poseedores de tierras de vocación forestal o agroforestal—PINPEP [84], provides small holder farmers in Huehuetenango with the opportunity to access forest incentives for sustainable land use management and agro-forestry practices.

## 4. Findings from the Case Study

This section presents findings from interviewees' perceptions and focus groups discussions about factors that either strengthen or limit the transformative capacity of niche innovations emerging from family farming in rural Guatemala.

Interview subjects were asked about the role government, private actors, umbrella farmer organizations, and donors played in strengthening the transformative capacity of grassroots innovations. Interview subjects assessed the role of these institutions as weak, average, or strong in relation to the role they play in various components and sub-components of transformative capacity, including agency interactions (C1–C3), core development processes (C4–C7), reflexivity, and social learning (C8) as well as relational representation forms (C9–C10) Table 5.

According to the interpretation of interviews and focus groups discussions, the umbrella farmer organization is considered the main catalyst of transformative capacity of grassroots innovations in rural Huehuetenango. The umbrella farmer organization is perceived as strong in terms of agency interactions (C1–C3), core development processes (C4–C7), and forms of relational representation (C9–C10). The umbrella organization is perceived as average in terms of reflexivity and social learning.

### 4.1. Regarding Agency and Interaction (C1–C3)

It includes participation and inclusion of different stakeholders. In this regard, the umbrella organization (ASOCUCH) engages the most vulnerable, such as women and youths, either in governance bodies inside the organizations, or in collaborative networks at municipal, department, and national levels, as well as in private–public alliances such as the honey and potato value chains.

- Inclusive and multi-forms of governance (C1)

Farmers, women, and youths also participate in diverse nodes and working networks (C1.2) both formal, informal, and at different levels: local, national, and international. This includes networks such as the smallholders' forest networks, municipal coordination groups such as Comisión de Fomento Económico, turismo Ambiente y Recursos Naturales (COFETARN), climate round tables, food security technical coordination groups at municipal and department levels, and women's municipal office. In this way, social and policy-advocacy capacities among member organizations are strengthened. The level of agency is strong, based on local ownership, and complemented by collaborative networks, round tables, and alliances at all levels.

> *"We need to work with the municipalities and government institutions, they are the normative entities, and we cannot work outside government's norms and regulations",* but we continue innovating and advocating for transformations in favour of family farmers and to preserve the surrounding ecosystems."* (Interview with value chain representative.)

The umbrella farmers' organization plays a crucial role as intermediary in promoting grassroots innovations and sustainability transformations. One of the key factors that legitimates and enhances the transformative capacity of ASOCUCH is its own governance, which is legitimate, inclusive, and intergenerational. This was highly appreciated by most interviewed subjects. Member organizations are elected in Assemblies with representation of all associations and cooperatives. Member organizations participate in decision-making as part of the board of directors. Representatives of farmers, women, and youth from members organizations participate actively in different commissions and working groups at the district, municipal, and national level. Knowledge and capacity-building on advocacy issues, leadership and gender equality and intergenerational perspectives is continuous and embedded in ASOCUCH's strategic plan of and in the action plans of its member organizations.

**Table 5.** Different stakeholders' role in building the transformative capacity of grassroots innovations.

| Components | Government (Go) | Private Sector (Pri) | Umbrella Farmer Organization (UFO) | Donors-NGO (Do) |
|---|---|---|---|---|
| | Weak (W)—Average (A)—Strong (S) | | | |
| **AGENCY INTERACTIONS FORMS (C1–C3)** | | | | |
| **Inclusive governance and multi-level governance (C1)** | Weak | Weak | **Strong** | Weak |
| C1.1 Participation and inclusiveness | W | W | S | W |
| C1.2 Diverse governance and network forms | W | W | A | W |
| C1.3 Sustained intermediaries and hybridization | W | W | S | W |
| **Transformative leadership and entrepreneurship (C2)** | Weak | Average | **Strong** | Weak |
| **Empowered and autonomous communities of practice (C3)** | Weak | Weak | **Strong** | **Strong** |
| C3.1 Addressing social needs and motives | W | W | S | S |
| C3.2 Community empowerment and autonomy | W | W | S | S |
| **CORE CAPACITY DEVELOPMENT PROCESSES (C4–C7)** | | | | |
| **System awareness and memory (C4)** | Weak | Weak | **Average** | **Strong** |
| C4.1 Baseline analysis and system (s) awareness | W | W | S | S |
| C4.2 Recognition of path dependencies | W | W | W | S |
| **Sustainability foresight (C5)** | Weak | Weak | Average | Average |
| C5.1 Diversity and transdisciplinary co-production of knowledge | W | W | W | A |
| C5.2 Collective vision for radical sustainability changes | W | W | A | A |
| C5.3 Alternative scenarios and future pathways | W | W | A | A |
| **Diverse community-based experimentation with disruptive solutions (C6)** | Weak | Weak | **Strong** | Weak |
| **Innovations embedding and coupling (C7).** | Weak | Weak | Average | Weak |
| C7.1 Access to resources for capacity development | W | W | A | W |
| C7.2 Planning and integrated transformative action | W | W | A | W |
| C7.3 Reflexive and supportive regulatory frameworks | W | W | A | W |
| **REFLEXIVITY and SOCIAL LEARNING (C8)** | Weak | Weak | Weak | Average |
| **Transformative knowledge** | W | W | W | A |
| Capacity for learning and monitoring | W | W | W | A |
| **RELATIONAL REPRESENTATIONS (C9–10)** | | | | |
| Different forms of agency (individual, households, institutions)—C9 | Strong | Weak | **Strong** | Average |
| Different scales (local, regional, national)—C10 | Strong | Weak | **Strong** | Average |

Source: own elaboration based on interviews and focus group discussions.

*"Changing the organizational culture is not an easy task. We have been working for many years in promoting women participation in the boards of local organizations; slowly women have gained some spaces, but it is hard for women to get higher positions, such as president and treasurer. Those positions are captured by experienced male leaders. We continue working with the boards in local farmers organizations to deal with issues of governance and gender. Some changes are happening, but not as faster as we wish, this takes time".* (Interview to a gender expert.)

As an umbrella farmers' organization, ASOCUCH allows the presence of intermediaries and hybridization (C1.3) through a strategy in which technical staff work together with farmer promoters and leaders. Often two people (farmer and technical staff) participate in different technical working groups, alliances, and round tables, in this way they address institutional, technical, and advocacy gaps. Constant capacity-building is conceived by technicians and farmers as an important condition to survive in the long-run. The inclusion of farmers gives legitimacy to their agendas; they can speak for the people they represent.

*"Some of the farmer-leaders from the farmer organizations are at the same level as experienced professionals when it comes to knowledge about seeds and climate effects in their territories. The only thing they are missing is a formal certificate, but the knowledge and know-how are superior to the theoretical formal education of newly graduated students."* (Focus group with technical experts from farmer organizations.)

The umbrella organization is engaged in multiple networks and collaborative alliances, involving public and private stakeholders. The umbrella farmers' organization understands the value of working with both decision-makers and technical staff in institutions, e.g., Ministry of Agriculture (MAGA), the National Forest Institution—Instituto Nacional de Bosques (INAB). ASOCUCH aims to improve food security and livelihoods of its members through strategic innovations that enhance livelihoods and resilient food systems.

*"Working for improved livelihoods is the way farmers' organizations can unite and share common visions to build resilient food systems."* (Interview technical expert in umbrella organization.)

- Transformative leadership (C2)

This was perceived by most interviewed people as strong in the umbrella farmers' organization. Leaders have appropriate skills. Not only senior leaders, but also new generations of boys and girls are prepared and participate actively in youth and gender commissions at district, department, and national levels. A common challenge is to avoid elite capture by senior leaders. Elite capture in terms of leadership is an issue that requires constant follow up, which sometimes creates divisions in communities. Occasionally, the board of directors in the umbrella farmer organization mediates if a conflict is not solved internally. Interviewees among farmer organizations expressed their lack of trust in municipal leaderships.

*"Leaders that are well-trained and are politically strong are really needed. The dilemma is that some of them want to stay in their positions forever, limiting the opportunity for others to get involved. Local farmer organizations must invest in youths and women. But this needs the understanding of old leadership and changes in the organizational culture to become more inclusive. Constant training is needed, we do not mean having a seminar, but the inclusion of these issues as part of the operative plans in the local farmers' organizations. Donors and the umbrella farmers' organizations should work towards getting the resources and capabilities needed."* (Focus group 1.)

*"We participate in activities and can get some money to work with farming activities. We also participate in some discussions in our organizations, but we do not feel that all leaders are happy when we women take leading positions. We work hard to gain our spaces. We are not used to speaking loudly, but men do. So, they win the discussions. Thanks to the exchanges promoted in CAV we feel that we are getting good opportunities*

> *to participate, but it is better when we have conservations with other women's farmers like us."* (Focus group 2.)

- Empowered organizations, networks, alliances, communities of practice (C3)

This includes responding to social needs (C3.1), which is a core activity for the umbrella farmers' organization. The importance of conserving the "micro-watershed" to sustain local livelihoods is understood by farmers' organizations. Conserving diversity and building resilience in family farming are common visions, which farmer organizations aim to upscale at district, department, and national levels. The umbrella farmers' organization works constantly to mainstream local demands and political agendas into national processes, new regulations, and initiatives.

> *"Some donors come with approaches that do not necessarily are adapted to the realities of our member organizations, or that we know have not worked previously. We must negotiate with donors, always trying to use our knowledge about what works and what doesn't work. Sometimes we have to say no to some interventions that our farmers organizations do not see as appropriate."* (Interview to technical expert.)

> *"It was good that CAV said that they needed women in the steering committees. Otherwise, we do not think we will have had the opportunity to participate and to equal access to credits."* (Focus group 2.)

Community empowerment and autonomy (C3.2): the members from the 20 associations and farmer cooperatives belonging to the umbrella organization become a community of practice in which innovations are tested and mainstreamed across households and local institutions. Innovations are promoted by the umbrella organization with the support of international donors, and through alliances with government institutions and private sector actors. The umbrella farmers' organization is the main responsible for the technical assistance, capacity-building, and mobilization of farmers for political advocacy and technical assistance. Leaders, local promoters, women, and youth receive training on technical issues, governance, and advocacy. Each organization has its own strategic and action plans; ASOCUCH brings innovation, capacity-building and pulls in resources from donors, private and government initiatives. Economic and technical resources are downscaled to the member organizations. Local promoters in the umbrella organization are constantly trained so they can provide quality extension services to their members. The autonomy of the umbrella organization is based on a combination of different initiatives. Representatives from ASOCUCH highlighted the importance of having resources for longer periods of time.

> *"Having allies among donors with a long-term perspective allows us to build up long-term strategies for sustainability and for the autonomy of our member organizations."* (Interview with a leader in farmer organization.)

*4.2. Critical Capacity Development Processes (C4–C7)*

- System awareness and memory (C4)

The micro-watershed as a unit of planning in CAV gives a more holistic approach within the territory. Baseline analysis and system (s) awareness (4.1). The vulnerability assessments in CAV are the departing point to build up the adaptation plan since it provides a baseline and the micro-watershed as unit of planning where the agri-food system is linked to water and forest ecosystems. In most communities, forest incentives complement activities and resources required to implement the adaptation plans designed in CAV. According to interviewed people, government institutions working with agri-food systems in Huehuetenango are still planning within traditional political boundaries. However, planning authorities at the department level see the advantages of CAV in creating linkages between different communities.

> *"Micro-watershed as a unit of planning is not new in Guatemala, but what is innovative from CAV is how communities work together to implement an adaptation plan that*

*goes beyond political boundaries. The ownership of the communities is notorious."*—(Interview with a donor.)

- Rural sustainability foresight (C5)

Regarding collective vision for radical sustainability changes (C5.1), perceptions from interviews and focus groups discussions support the existence of a sustainability vision in rural districts of Huehuetenango, but there is no consensus on which is the common vision or what the resources required to achieve it are. Most institutions still operate in silos.

Interviewees argued there is a shared vision to build resilient agri-food systems in rural Huehuetenango, although pathways to get there are still uncoordinated. Many of those interviewed argued that most development actors work in silos with their own targets' groups and approaches. ASOCUCH has tried to align their strategic priorities to those of Kat'un 2032, but their achievements are still not reported as part of the achievements at the municipal level.

Diversity and transdisciplinary production of knowledge (C5.2): Interviewees argued that the Guatemalan government is committed to work towards resilient agri-food systems, as defined in Kat'un 2032, but they also expressed concerns about the lack of extension services from government institutions in rural Huehuetenango. District and department level authorities prioritise infrastructure projects in detriment of investments on resilient production with smallholders' farmers. Government institutions such as the Ministry of Agriculture (MAGA) have been working towards the rehabilitation of Rural Development Learning Centres (CADERS) by training extension agents on soil conservation, water management, horticulture production, food security, and nutrition; so far 10 extensionists were trained in Huehuetenango in 2019. These are positive steps, but MAGA still has limited capacity to reach all farmers. Government institutions in rural districts are working in alliances with umbrella farmers' organizations such as ASOCUCH, to reach more farmers with their strategies and plans.

*"We coordinate some actions but are not aligned to a common objective or to the Kat'un to give an example. However, I must say that the umbrella farmer organization is creating a certain level of coordination. They are doing it through the different round tables they have organized, such as the one for forest incentives, and the climate round table. We are now leading the climate round table, but the umbrella organization created it and gave life to it in the beginning and for many years. So, this second-tier farmers' organization is a driving force behind these initiatives. It is very important for us to have such a partner in Huehuetenango."* (Interview with a donor's representative.)

Alternative scenarios and future pathways (C5.3): climate adapted villages (CAV) bring innovative pathways towards more resilient agri-food systems in rural Huehuetenango. This vision is embedded in strategies of the umbrella organization and action plans in the member organizations. According to the interviews, responding to basic social needs is always the departing point. It was highlighted that the umbrella farmers' organization plays a key role in promoting innovations and transitions towards resilient agri-food systems in rural districts of Huehuetenango.

*"CAV as an approach was developed in a participatory way with the donor. We (the umbrella organization) got involved in discussions with the donor from the beginning in order to tailor the approach to our needs. This perhaps explains the success of CAV. The entry point for CAV was not the environment or climate, but rather it came as a solution to economic and social needs in the communities. In the process, CAV created understanding among community members on the importance of conserving the natural resources in the micro-watershed. It also increased collaboration among communities that traditionally did not work together."* (Interview with technical staff in UFO.)

- Diverse community-based experimentation with disruptive solutions (C6)

Interviewed technical staff and directors at ASOCUCH expressed the importance of having constant innovation as part of the progress and long-term permanence of the

organization. Highly qualified technical staff in ASOCUCH and local farmer promoters trained in the member organizations are developing new ideas, which are rooted in local needs and realities. The umbrella farmers' organization brings innovations to farmers, either by organising the resources from government incentives for forest protection, by mobilising own resources (micro finance systems), or by advocating for the allocation of more resources from the municipal budgets to smallholder farmers.

> *"Our approaches are participatory and bottom- up. We understand that strong leaders are instrumental to get the changes needed. We can support them technically, but they need to frame their demands directly. We technicians are there to support, but local farmers are the ones that need to face government institutions and politicians. For example, every year they are negotiating in the National Congress to demand an annual budget for forest incentives for smallholders. It is a struggle every year, and they negotiate by themselves. Years of training are needed. You do not reach to that level of negotiation in one day, you need years of preparation."* (Focus group 1.)

ASOCUCH sets the agenda with innovative solutions, such as the climate adapted villages (CAV), which becomes self-sustained by using green micro-credits. Green micro-credits are schemes managed by a local farmer cooperative that require environmental improvements such as soil conservation practices, agrobiodiversity conservation, and agro-forestry alternatives as an additional repayment of the loans. Despite budget limitations, ASOCUCH has implemented CAV in 11 micro-watersheds and 72 communities. The replication of CAV depends on availability of funds from donors, or via the incorporation of compensations schemes for agro-forestry and any other system that builds resilience in rural agri-food systems.

- Innovation embedding (C7)

The integration of CAV into existing funding schemes, such as the PINPEP is still an ongoing process. However, CAV is part of the routines and strategies among farmers organizations, and one of the main approaches promoted by the umbrella farmers' organizations among donors and government institutions. With the support of diverse donors, CAV has been implemented and adopted by 72 farming communities and 11 different micro watersheds. The umbrella organization has been instrumental in capacity development and in providing basic resources to launch CAV among rural communities in Huehuetenango, Guatemala. CAV has also been adopted by other farmers organizations working in different departments of Guatemala.

Access to resources for capacity development (C7.1): ASOCUCH promotes novel ideas and remove barriers to grassroots innovations. Diverse donors have been engaged, and a possible coupling strategy is to link CAV to existing programs, e.g., forestry incentives for smallholders or "Programa de incentivos para poseedores de pequeñas extensiones de tierras de vocación forestal o agroforestal" (PINPEB). PINPEB is key to all member organizations in ASOCUCH, since it is one of the regime changes that emerged from their own advocacy. This scheme was created as a response to the demands of smallholders who did not benefit from previous forestry incentives. Getting annual budget allocation for PINPEB is a struggle every single year, and farmers from ASOCUCH and the country must mobilize in the Congress for the approval of budget allocation for PINPEB.

> *"It is not easy to create new incentive systems, because it demands long-term processes, new legislation. There has not been a single year without us advocating to Congress to get budget allocations for PINPEB. But we already have PINPEB, and it can be strengthened with resilient agriculture. It is also helps to get budget allocations in the Congress."*—(Interview with technical coordinator in umbrella organization.)

Planning and mainstreaming transformative actions (C7.2): strategies and plans are created to reduce barriers to innovation. Results from innovations are used for policy advocacy and to create discourses and advocacy agendas, which can allow member organizations to mainstream innovations and keep innovating.

*"Innovations such as CAV are included as a core approach in our strategic plan (UFO), then each farmer organization can include it in their operative plans. This means that they must discuss it in the boards and get to a common understanding that CAV is something they want to have. We promote exchanges between those farmer organizations that have CAV and those who do not have it. This helps in the understanding of CAV's potential to solve local needs. Resources for innovation are limited, so we cannot experiment a lot, but if it is something that works for farmers, they will get involved"* (Interview with staff in the Umbrella organization.)

Reflexive and supportive regulatory framework (C7.3): PINPEB allows smallholders to have permanent incentives to protect forest and agro-forestry systems. Instead of creating new laws, some of the interviewees argued it is better to expand the scope of the PINPEB to strengthen the conservation and diversification efforts in agro-forestry systems and in that way contribute to building resilient agri-food systems.

### 4.3. Reflexivity and Social Learning (C8)

Learning and reflexivity was perceived as weak by most interviewed stakeholders. Farmers' organizations want more reflection about the way forward with CAV. They also expressed that government institutions and most donors have a short-term permanence, which hinders reflexivity and social learning. Limitations in terms of permanent staff in government institutions makes it difficult to plan and to collaborate in a long-term perspective. It was observed that there is no institutional memory; new employees start their work with almost no documented information from their predecessors. This is difficult for those institutions and donors struggling with high personnel turnover and short-term funding cycles of less than two years. The umbrella organization has few donors committing funds for 5-year periods and it helps in getting long-term changes. Several participants in interviews and focus group discussions expressed that those changes in people's mindset require long-term commitment. Representatives from the umbrella farmers' organization argued that learning is a strong factor to move innovations forward. Highly qualified farmers get all technical skills to be able to do advocacy at any level of agency. Some local adaptation committees need preparation in advocacy issues, especially those living in distant districts.

*"We at the umbrella organization prepare local technicians who oversee the technical assistance within each local farmer organization. Some farmers organizations are paying their local technician 100%, but most of them still need support to get to the level where they can afford the annual salary of a local technician. Ideally, they should have more spaces for learning and reflection, and we do understand that. One way we can do that is through organizing round tables (networks) around some topics of common interest, such as putting together those working in seeds and forest incentives in round tables to discuss issues of relevance, both for technical and advocacy purposes. We encourage donors to support capacity development that addresses the organizational culture: this can lead to more reflection and learning among farmers organizations. However, very few donors can commit to this because a majority of donors are pressed to get results in the short run, leaving almost no time to reflexivity and learning."* (Interview with technical staff in UFO.)

### 4.4. Relations at Different Levels of Agency and Scales (C9–C10)

The umbrella farmer organization ASOCUCH oversees niche innovations, mobilises human and financial resources, and builds alliances for critical capacity development processes towards sustainable agri-food systems in rural districts of Huehuetenango.

- Different levels of agency (C9)

ASOCUCH facilitates transformations at different levels of agency: individuals, households, farmers' organizations, institutions, and in multi-institutional collaborative alliances. The interaction at individual, local, national, and international scales is considered average

by interviewed subjects. Most government agencies such as the Ministry of Agriculture (MAGA), the National Protected Areas Council—-Consejo Nacional de Areas Protegidas (CONAP)—-and some international non-governmental organizations transit between different scales, from district to national level. There are some gaps in terms of technical assistance for resilient agriculture among most government institutions.

According to interviewees, most stakeholders in rural Huehuetenango have a "silo approach", in which institutional priorities weigh more than common sustainability agendas. Multi-institutional collaboration is an emerging arena for many government institutions and even donors. There are some positive steps, e.g., the honey public–private collaborative initiative.

> *"The umbrella organization has the ability to coordinate actions in the field. There have been numerous dialogues with us that represent the donors to channel our funding to initiatives such as CAV. The umbrella organization is constant, has a long-term presence in the territory and they are respected and trusted by their organizations, but also by us working in development. This respect is the result of having clarity on solving the needs of their members' organizations."* (Focus group 1.)

ASOCUCH as a farmers' network facilitates different forms of agency and interactions in rural Huehuetenango. It is perceived as a catalyst organization in Huehuetenango by their member organizations and relevant government institutions, e.g., Ministry of Agriculture (MAGA), Instituto Nacional Forestal (INAB), Consejo Nacional de Areas Protegidas (CONAP), by donors, and other non-governmental organizations and by Municipal authorities.

- Different scales (C10)

ASOCUCH's member organizations are part of different coordinating groups, such as the Alliance of beneficiaries of PINPEB and round tables on climate change, at department and at national level. ASOCUCH is also a member of COFETARN "Comisión de Fomento Económico, Turismo, Ambiente y Recursos Naturales" at the municipal level. COFETARN is formed by relevant development stakeholders that have a presence in the districts and municipalities in Huehuetenango.

> *"The collaborative alliances and round tables organized by (the umbrella organization) are facilitating technical collaboration, donor alignment, but also creating appropriate arenas to discuss and join advocacy efforts to achieve changes for family farmers in Huehuetenango"* (Focus group 1.)

> *Gender gaps*: most women and men interviewed argued that women-led agriculture is not a priority for a district's level funding. Local facilitators and coordinators related to the implementation of women initiatives expressed they have limited influence in decision-making, especially at the district level. There are national initiatives targeting women, but these initiatives are not prioritised for budget allocations by district level authorities.

> *"We women are not prioritized by the municipalities; there is not money for women's projects. Our local organizations and the umbrella organization bring us opportunities such as CAV. CAV provides us loans and trainings. Otherwise, we are left out."* (Focus group 2.)

> *"It is difficult to gain spaces as women in rural settings, CAV has helped us to become part of the steering committees. In the beginning we did not feel capable, but the trainings received from the umbrella organization helped us to get the courage and trust in ourselves that was needed to do good work."* (Focus group 2.)

Interviewed subjects added that more advocacy is required at all levels of agency to ensure that women are targeted not only in plans, but in innovation initiatives, budget allocations, and investments.

> *"Requesting women's participation in decision-making bodies in CAV, in the planning and in affirmative actions, was something that we as donors requested. It was contested*

*at the beginning by male leaders in the grassroots organizations, but they understood that it was needed in order to get CAV in their communities. But there is still a long way to go in this field."* (Interview with a donor.)

## 5. Discussion: Enabling and Limiting Factors

In this Section 5, we discuss and group findings about enabling and limiting factors towards transformative capacity in the case of Huehuetenango. Evidence supporting this analysis comes from interviews and focus groups as presented in Section 4 about findings.

### 5.1. Enabling Factors

The existence of an umbrella farmer's organization with a long-term commitment plays a key role in building transformative capacity of "niche" grassroots innovations. The umbrella farmer organization plays different bridging roles, such as network building, building a common sustainability vision, and becoming an intermediary in promoting grassroots innovations. This helps to move CAV from niche to broad implementation by different farmers and communities. The transformation is not radical, but enough to guarantee better living conditions of farmers and the resilience of local agri-food systems.

In our case study, the umbrella organization ASOCUCH is the catalysing factor that enhances the transformative capacity of rural Huehuetenango. Some elements that nurture this are its ability to enable different interaction forms (C1–C3) in which the following factors are crucial:

Responding to economic and social needs first: the umbrella farmer organization prioritises opportune solutions to economic and social needs among its member organizations. Environmental protection is important because it is the basis for local livelihoods. Understanding this order is crucial when designing paths towards resilience in agri-food systems in rural Huehuetenango as expressed by technical experts.

Creating local technical competencies: strong and opportune technical backstopping in the umbrella headquarters, is supported by highly qualified local promoters in member farmer organizations. Continuous capacity building to farmers, youths, women leaders, and local promoters is part of the success in the adoption and up-scaling of grassroots innovations promoted by the umbrella organization, contributing in this way to creating empowered communities of practice.

Legitimate participation: farmer representatives participate in round tables, local bodies, e.g., COFETARN. Members participate with legitimate representation. They bring their own voices to the different fora, and they are well-prepared.

The study also shows that the umbrella organization facilitates and provides capacity-building to farmers and even other stakeholders on key capacity development processes (C4–C7). Perceptions from both interviews and focus groups discussions.

Local ownership of common resilience visions: seeking ownership among the member partner organizations on common visions and strategies towards resilience in local agri-food systems is an element of success for the umbrella farmer organization. Such resilience vision is nurtured by capacity-building on technical and advocacy issues that is tailored and owned by member farmers' organizations. The network of local promoters from all member organizations ensures the extension services needed and the innovation required, contributing in this way to creating diverse community-based experimentation with disruptive solutions.

Baseline analysis and system (s) awareness and the recognition of path dependencies: the umbrella organization through innovations such as CAV has created systems awareness, as well as the recognition of path dependencies when building resilient agri-food systems in rural Huehuetenango. CAV is helping the communities to see the local agri-food systems at the micro-watershed level, instead of focusing on households.

The umbrella farmers' organization and its extended collaborative networks and alliances are pivotal in building relational dimensions (C9–C10) by creating changes at different levels of human agency: individual, households, and communities, and at different

scales. It provides technical assistance to farmers and navigates across different levels of agency (households, communities, networks, and institutions) and across different interactions scales (local, department, and national). One of the key success factors is the work in alliances and round tables.

Collaborative alliances and networking: The umbrella organization brings legitimacy and the voices of male and female farmersand youth to different round tables and policy-advocacy from local to national level. The umbrella farmers' organization and its representation in diverse technical and advocacy networks enables the replication of communities of practice and experimentations around grassroots innovations that otherwise would not reach family farmers in rural Guatemala.

Mobilisation of resources: Grassroots innovations such as CAV are promoted among all member organizations who can experiment and share learning and experiences. The umbrella farmers' organization mobilises resources and alliances for critical capacity development processes and advocates for funding at the national and district level.

*5.2. Limiting Factors*

As we have illustrated in our case study, the catalyst ability of the umbrella organization is constrained by the following factors:

Elite capture of senior leaders: multi-generational transitions are still restrained by some leaders in the member farmers' organizations. Opening spaces for young generations is challenging in some of the member organizations. Gender and multigenerational gaps: the umbrella organization is still facing gender and multi-generational gaps in some of the member organizations. Although some quotas were required in CAV, women and youth's participation in innovation requires some extra attention as expressed by many interviewed people.

Resources for capacity development are scattered: Capacity-building in the umbrella organization is still based on donor support. The umbrella organization is building a system based on self-sufficient extension services entirely supported by the member organizations. The micro-credit funds in CAV help to maintain capacity-building activities. However, more capacity building is required for both technical and advocacy issues.

Reflexivity and social learning: effective advocacy at local level requires constant reflexivity and learning in member organizations, this still depends on the umbrella organization. Transformative knowledge needs reinforcement, as expressed by interviewed stakeholders.

Short-term perspectives in donor funding: our analysis has clearly shown how most international donors and government institutions in Huehuetenango, Guatemala, have a short-term approach towards building resilient agri-food systems. At the same time, enhancing the transformative capacity of grassroots innovations requires a long-term perspective to become embedded in family farmers and in local and national institutions. A short-term perspective of funding for capacity building and innovations hinders the transformative capacity of grassroots innovations to overcome niche status, as mentioned by interviewed stakeholders.

Our case study shows that decision-makers at district level give priority to investments in infrastructure projects, e.g., roads, in detriment of investments in family farming and local agri-food systems. Our analysis also shows weaknesses in government extension services, as well as scattered engagement from government and research institutions in research and innovation in rural districts of Huehuetenango.

The umbrella farmer organization enables different forms of interaction by responding to economic and social needs first, and by enhancing technical and advocacy competencies to ensure legitimate representation. It mobilises resources and alliances for critical capacity development processes and advocate for funding to incentives' schemes at national and district levels. It provides technical assistance to farmers and navigates across different levels of agency (households, communities, networks, and institutions) and across different scales of interaction (local, department and national).

## 6. Conclusions

The analogy with Wolfram's [53] framework was useful for the analysis of factors enabling and limiting the transformative capacity of grassroots innovations towards resilient farming systems in rural Huehetenango, Guatemala.

The existence of a legitimate and inclusive umbrella farmers' organization is a crucial factor that strengthens the transformative capacity of grassroots innovation in rural family farming systems in Huehuetenango. The representation of the umbrella farmers' organization in diverse technical and advocacy networks enables the replication of communities of practices and experiments around grassroots innovations, which otherwise would not reach family farmers in rural Guatemala.

The umbrella farmers' organization plays a catalyst role in bringing innovations and technical assistance to farmers, in promoting transitions to resilient farming systems at different levels of agency from individuals, households to local institutions, and across different interactions scales: local, national, and even international. The umbrella farmers' organization is powerful and has a strong agency role, which is crucial to overcome niche status of grassroots innovations

Scattered, and short-term funding for capacity-building and social learning in farmer organizations also limit the transformative capacity of grassroots innovations.

Gender and multi-generational gaps are major factors that limit the transformative capacity of grassroots innovations in rural Guatemala and deserve further research.

**Author Contributions:** Conceptualization, R.O. and J.P.; methodology, R.O. and J.P.; validation, R.O. and J.P. formal analysis, R.O.; investigation, R.O.; writing—original draft preparation, R.O.; writing—review and editing, R.O. and J.P.; supervision, J.P.; project administration, R.O. All authors have read and agreed to the published version of the manuscript.

**Funding:** This research received no external funding.

**Institutional Review Board Statement:** Not applicable.

**Informed Consent Statement:** Informed consent was obtained from all subjects involved in the study.

**Data Availability Statement:** Not applicable.

**Conflicts of Interest:** The authors declare no conflict of interest.

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
