# Peer review of "The Role of Farmers’ Umbrella Organizations in Building Transformative Capacity around Grassroots Innovations in Rural Agri-Food Systems in Guatemala"

_sustainability, doi:10.3390/su14052695_

Round 1

Reviewer 1 Report

Previous comments are not well addressed in this version. Specific comments are as below.

[1] In line 6-7, only authors’ affiliations are needed here.

[2] In line 10-26, usually abstract is a paragraph containing the main idea, data, methodology and results.

[3] In line 33, [16][17][18] can be like [16-18]. Lots of same cases appear in the whole paper.

[4] In line 230 “-adapted from Wolfram [15],(page 126)”, how many texts are from Wolfram’s work? What are authors’ important work?

[5] In line 233-234, this sentence is hard to understand. It looks like all citations in this paper needs to be improved.

[6] In line 98, figure title should be under figure. The same case appears in line 138.

[7] In line 97 and line 114, why is “Figure 1.” here?

[8] In line 130 “(page 126)”, I don’t think page number is needed.

[9] In line 137, where is “Figure e2”?

[10] Section 2.1.1 to 2.1.4 looks like all come from others’ work. What is the main idea of this study?

[11] In line 301, why “Map 1” appears again?  

[12] This paper looks like a report, not a research article. It contains too many texts from others work. It is not well structured, and the writing looks not good, especially in abstract, citation, findings and discussion. Writing skill needs to be improved. I still don’t think this paper is present in good quality and is ready for publishing.

Author Response

Kind regards,

Reviewer 2 Report

The aim of the paper is to study the role of farmer umbrella organizations in building and developing tranformative capacity in order to scale-up sustainable and resilient agro-food transition. The study presents a case study which is about a grassroots innovation called „Climate adapted villages” promoted by ASOCUCH as an umbrella organization situated in Huehuetenango, Guatemala.

The study proved that umbrella organizations play a significant role in increasing transformative capacity of local actors and therefore enabling them to pursue socio-technical transformations. The case study reveals the enabling and limiting factors of the investigated umbrella organization which are greatly influence the catalytic role of this organization. Consequently, research contribution in the paper can be identified, so, the results provide an advance in current knowledge. The study provides new insights and new knowledge on the topic in question.

The topic covered in this study is absolute relevant and so important in terms of sustainable rural development. Enabling and empowering rural actors to be active participants in shaping their own destiny requires great attention and appropriate actions on the part of those concerned. The article correctly points out the factors that help and inhibit transformative ability of local rural communities. In this way the study provides useful information and knowledge on how to create and develop more sustainable and resilient communities by generating novel bottom-up solutions, community initiatives for sustainable development trough networks of local and extra-local actors.

The title is in line with the content.

The abstract is lacking of results, conclusion, and implications of findings. It should be reworked accordingly.

In the first chapter the authors briefly introduce the topicality and relevance of the study, and formulate a research goal based on the identified research gap. The structure, list of chapters of the study is also provided in this chapter.

The second chapter provides detailed information on the multi-level perspective (MLP) and transformative capacity framework. In this chapter the authors provide theoretical foundations for the analysis using appropriate references.

The third chapter presents the methods used in the research. Interviews and focus group discussion as qualitative methods were applied which are widely used in the social sciences and are suitable for achieving the research goal. The case study was also presented in this chapter. The specific subject of the study is the “Climate adapted villages” initiative which is a grassroots innovation emerged in the area of Huehuetenango, Guatemala, and fostered by ASOCUCH umbrella organization.

Chapter four is presenting the findings of the research. The results are shown and discussed in detail regarding the transformative capacity framework. This chapter is well based, well presented and well explained. With the presented case study the paper provides better insight into the topic in question.  

The discussion chapter provides a presentation and explanation of enabling and limiting factors of the examined umbrella organization in building transformative capacity, and makes clear the added value of the study to the existing knowledge and points out the theoretical contributions, which is confirmed by the conclusion.

Specific comments

In line 58: „may” instead of „my”

In line 72-74: Andersen, et al – correctly: ; Andersen et al.; Avelino et, al – correctly: Avelino et al. (this type of error occurs elsewhere)

In line 137: „regional” instead of „regions”

In line 210: „knowledge creatin” - knowledge creation

In line 216: „theprocess” – the process

In line 224: „The set of components explained in 2.2 were” – there is no 2.2 section in the text

In line 253: „the first list of interviewers” – interviewees

In Table 1: Under the designation „Government (Go)” there are FO1 and FO2, but they are not governement agencies; they are rather producer organizations

In line 272: „Huehuetenango's is one of the most diverse in terms of Mayan ethnic groups.” - Huehuetenango's is one of the most diverse REGION in terms of Mayan ethnic groups

In line 367: „interviewers’ perceptions” - interviewees’perceptions

In line 446: „Interviewers” – interviewees

In line 585: „ehey” - They

Author Response

Please see Attachment.

Kind regards,

Reviewer 3 Report

The manuscript "The role of farmer's umbrella organizations in building transformative capacity around grassroot innovations in rural agri-food systems in Guatemala" deals with a very interesting topic, which is well suited within the aims and scopes of Sustainability journal. However, I have some concerns regarding the methodological framework and approaches used to answer the formulated research question.

First, the transformative capacity framework developed used has been developed for urban context. The authors must how they adapted it to be used in the rural contexts.

Second, the authors seeks to assess the role of umbrella farm organisations in building transformative capacity by using one organisation as case study. The answer to this question is provided in the 1st part of the study and reported in Table 2.  However, as metioned, the case study organisation is not only engaged in fostering the Climate Adapted Villages (CAV) programme. Therefore, the authors must explain how they disentangled the general role of the organisation in forstering innovations from its role in promoting the CAV grassroot innovation.

Third, in the Findings session, the authors describe the stengthening and limiting factor for transformative capacity in rural Guatemala following the components of Wolfram framework. The authors focused manly on one case study umbreall organisation describing the role it plays in fostering (limiting) transformative capacity based on the perceptions of interviewees and focus group participant, only. Evidences supporting these perceptions must be provided. 

Other remarks:

  • The number of stakeholders and their categories participating in the focus group must be reported.
  • Lines 248-252 "Document analysis": A list of the analysed documents must be provided in table format.
  • Table 2: FO1 and FO2?
  • Line 129: "path-deviant" what?
  • the manuscript has several typos, english language must be revised.

Author Response

Please See Attachment.

Kind regards,

Round 2

Reviewer 1 Report

All previous comments are not well addressed in the latest version. The logical structure and writing quality of the whole paper are still not ready for publication. Specific comments are as bellows.

[1] There are extra spaces in line 88, 106 and 197 and extra full stop in line 85. Something is wrong in line 47 and 90. It looks like all citations need further format improvements.

[2] In line 95, why is “Figure 1.” here? Does it need brackets?

[3] In line 97 and 133, center all figure titles and align left “source” in a new line below figure title.   

[4] In line 257, 282, 293 and 308, table titles should be centered on top of tables. And align left “source” in a new line below table title.   

[5] In line 349, why is “Map 1.” still here?  “Map” is not a caption.  I suggest “figure 3” here.

[6] Section numbering needs improvement. Look at section 2, there is only section 2.1 including section 2.1.1, 2.1.2, 2.1.3 and 2.1.4. I don’t find section 2.2 or 2.3 in this paper.

[7] In line 433, where is the title of section 4? Any titles for C1 and C2 like that of C3 in line 535?

[8] In line 727, any titles for C9 and C10 like that of C3 in line 535?

[9] Align left titles of section 3.1.1, section 3.1.2 and section 3.1.3.

[10] For the interview details in section 4, are they necessary? There are too many texts from other’s work or interviews, which makes this paper looks like a report, not a research article. The authors should summarize previous work or interview details in own words rather than just citing all those texts in the paper.

Author Response

Response to Reviewer 1 Comments

All previous comments are not well addressed in the latest version. The logical structure and writing quality of the whole paper are still not ready for publication. Specific comments are as bellows.

We appreciate your comments and apologize for not addressing properly your comments in previous versions of the manuscript.  We have been improved the logical structure of the whole paper as per your recommendations, we hope this can respond to your comments in a more appropriate way. Response to concrete comments is provided under each of comment below. 

[1] There are extra spaces in line 88, 106 and 197 and extra full stop in line 85. Something is wrong in line 47 and 90. It looks like all citations need further format improvements.

Amended: extra spaces deleted in line 321.

Extra spaces have been deleted in different lines of the document.280,

[2] In line 95, why is “Figure 1.” here? Does it need brackets?

Amended: brackets have been added in line 121, which corresponds to new lines’ numeration in the last version of the manuscript with track changes.

[3] In line 97 and 133, center all figure titles and align left “source” in a new line below figure title.   

All titles are centered, and all sources are aligned to the left.

[4] In line 257, 282, 293 and 308, table titles should be centered on top of tables. And align left “source” in a new line below table title.   

Amended: table 1 title on top, reference is made to line 300 that corresponds to the new version of the manuscript with track changes.

Table 2 caption is placed above, reference is made to line 324 in new version of the manuscript with track changes.

Table 3 caption is placed above, reference is made to line 338 in new version of the manuscript with track changes.

Table 4 caption is placed above, reference is made to line 357 in new version of the manuscript with track changes.

Sources to all tables are aligned left below the tables (ref. lines 302, 329, 341, 361)

[5] In line 349, why is “Map 1.” still here?  “Map” is not a caption.  I suggest “figure 3” here.

Amended: Figure 3 is added as per your recommendation. Reference is made to line 404 in new version of the manuscript with track changes.  Reference is made to figure 3 in line 401 in brackets.

[6] Section numbering needs improvement. Look at section 2, there is only section 2.1 including section 2.1.1, 2.1.2, 2.1.3 and 2.1.4. I don’t find section 2.2 or 2.3 in this paper.

Numeration for section 2 have been improved according to your recommendation. Previous 2.1 has been deleted; 2.1.1. has been replaced by 2.1; 2.1.2 by 2.2; 2.1.3 by 2.3 and 2.1.4 by 2.4; the titles for this section have been slightly modified as follows:

2.1 Transformative capacity and agency interaction forms (C1-C3) -ref. line 174

2..2 Transformative capacity and core capacity development processes (C4 -C7)- ref. line 201

  1. 3 Transformative capacity and learning and reflexivity (C8)- ref. line 233

2.4 Transformative capacity and relational representations (C9-10) -ref. line 246

[7] In line 433, where is the title of section 4? Any titles for C1 and C2 like that of C3 in line 535?

The title of section 4 is findings that has been replaced by 4. Findings from the case study (line 467).

Inclusive and multi-forms of governance (C1)-line 497 and Transformative leadership (C2) in line 563 have been highlighted as titles as in C3 and the rest of the components.

[8] In line 727, any titles for C9 and C10 like that of C3 in line 535?,

Your recommendation is appreciated. Titles have been added for Different levels of agency (C9) in line 806 and Different scales (C10) line 835 corresponding to the new lines’ numeration in the new version of the manuscript with track changes.

[9] Align left titles of section 3.1.1, section 3.1.2 and section 3.1.3.

Titles for sections 3.1.1; 3.1.2 and 3.13 have been aligned to the left. Reference to lines 279 and 303 and 343.

[10] For the interview details in section 4, are they necessary? There are too many texts from other’s work or interviews, which makes this paper looks like a report, not a research article. The authors should summarize previous work or interview details in own words rather than just citing all those texts in the paper.

Additional evidence from interviews and focus groups discussions were requested by other reviewers, this is the reason why we included additional citations in section 4. About findings.  As part of the methodology in qualitative research it is important that findings reflect and bring into the writing the voices of interviewed people with their own quotations (section 4. about findings). in section 5. authors discuss the findings.

Date: 17.02.2022

Reviewer 2 Report

In my opinion, the manuscript has improved a lot, since the authors have taken all of the comments made by the reviewers into consideration, they modified and corrected the text accordingly.

I accept it in present form.

Author Response

Dear Reviewer

We appreciate your positive feedback regarding the content of the paper.

In response to your comment about the English edition, the manuscript has been subject to  a comprehensive English editing.

Kind regards,

Reviewer 3 Report

The authors answered to all my remarks. Thanks.

Author Response

Dear Reviewer

We appreciate your positive feedback regarding the content of the paper.

Kind regards,